# Production and Preliminary Characterization of Linseed Mucilage-Based Films Loaded with Cardamom (*Elettaria cardamomum*) and Copaiba (*Copaifera officinalis*)

**Mayra Z. Treviño-Garza** [1], **Ana Karen Saldívar-Vázquez** [2], **Sonia Martha López-Villarreal** [2], **María del Refugio Lara-Banda** [3], **Joel Horacio Elizondo-Luevano** [4], **Abelardo Chávez-Montes** [4], **Juan Gabriel Báez-González** [1,*] and **Osvelia Esmeralda Rodríguez-Luis** [2,*]

1 Departamento de Alimentos, Facultad de Ciencias Biológicas, Universidad Autónoma de Nuevo León (UANL), San Nicolás de los Garza 66455, NL, Mexico; mayra.trevinogrz@uanl.edu.mx
2 Departamento de Microbiología, Facultad de Odontología, Universidad Autónoma de Nuevo León (UANL), Monterrey 64460, NL, Mexico; ana.saldivarvz@uanl.edu.mx (A.K.S.-V.); sonia.lopezvl@uanl.edu.mx (S.M.L.-V.)
3 Centro de Investigación e Innovación en Ingeniería Aeronáutica, Facultad de Ingeniería Mecánica y Eléctrica, Universidad Autónoma de Nuevo León (UANL), San Nicolás de los Garza 66455, NL, Mexico; maria.laraba@uanl.edu.mx
4 Departamento de Química, Facultad de Ciencias Biológicas, Universidad Autónoma de Nuevo León (UANL), San Nicolás de los Garza 66455, NL, Mexico; joel.elizondolv@uanl.edu.mx (J.H.E.-L.); abelardo.chavezmn@uanl.edu.mx (A.C.-M.)
* Correspondence: juan.baezgn@uanl.edu.mx (J.G.B.-G.); osvelia.rodriguezls@uanl.edu.mx (O.E.R.-L.); Tel.: +52-8183294000 (ext. 3654) (J.G.B.-G.); +52-8183294230 (ext. 3117) (O.E.R.-L.)

**Abstract:** In this research, developed linseed mucilage (M)-based films loaded with *E. cardamom* (MCA), *C. officinalis* (MCO), and co-loaded with both compounds (MCACO) were evaluated. The incorporation of the active compounds modified the color (redness–greenness, and yellowness); however, the thickness remained constant in all treatments (0.0042–0.0052 mm). In addition, the solubilization time of the films (in artificial saliva) to release the active compounds fluctuates between 9 and 12 min. Furthermore, the incorporation of bioactive compounds increased the total phenolic content and antioxidant activity (DPPH and ABTS, respectively), mainly in MCA (inhibition of 81.99 and 95.80%, respectively) and MCACO (inhibition of 47.15% and 39.73%, respectively). In addition, the incorporation of these compounds also decreased the hardness (39.50%–70.81%), deformation (49.16%–78.30%), and fracturability (39.58%–82.95%). On the other hand, it did not modify the adhesiveness, except in MCO. Moreover, SEM micrographs showed a more homogeneous structure in the MCO films among the films that contained CA in the formulation (heterogeneous structure with the presence of protuberances). Finally, due to the previously reported pharmacological properties of *E. cardamomun* and *C. officinalis*, the films developed in this study could have an application as a wound dressing in dentistry.

**Keywords:** films; linseed mucilage; cardamom; copaiba; physicochemical properties; antioxidant activity; texture analysis; dentistry

## 1. Introduction

Nowadays, there is a growing global trend towards developing new natural products, such as therapeutic options for the prevention and treatment of various diseases [1,2]. Due to this, numerous researchers have determined the properties of multiple natural products with active properties of interest to the medical area [3]. In addition, recent research has focused on studying phytotherapeutic products of interest to dentistry [4–6].

Cardamom (*Elettaria cardamomum*; CA) is a perennial, aromatic, herbaceous plant belonging to the *Zingiberaceae* family, whose use is culinary, domestic, and medicinal. It is a

species native to southwest Asia (India, Sri Lanka, Malaysia, and Indonesia), Tanzania, and Guatemala, which has been used since ancient times due to its wide variety of antimicrobial, antifungal, antioxidant, analgesic, anti-inflammatory, and anticancer properties associated with the presence of components such as phenols, tannins, terpenoids, flavonoids, and sterols, among others [7–9]. Further, in dentistry, it has been reported that CA possesses anti-bad breath, anticaries, antiseptic, and antimicrobial properties [10]. Multiple studies have shown that CA has been effective against microorganisms that cause infections and dental cavities [11], such as *Streptococcus mutans, Candida albicans*, and *Lactobacillus casei* [12–15]. Additionally, it has been shown that the use of CA in combination with other extracts (black pepper/black cumin/cardamom and black pepper/black cumin/cardamom/cinnamon) has a high antibacterial susceptibility against microorganisms from oral isolates [16]. Finally, some reports indicated the potential therapeutic benefits of CA for periodontal infections [17].

Moreover, *C. officinalis* is a plant belonging to the *Fabaceae* family, distributed mainly in Latin America (Brazil, Bolivia, Colombia, Peru, and Venezuela) and West Africa [18–20]. The main uses of CO include medical applications, nutrition, cosmetics, fuels, and wood, among others [1]. CO has been widely used in folk medicine through topical and oral administration due to its antimicrobial, antifungal, antiseptic, anti-inflammatory, and antioxidant properties, among others [3,6,20]. Regarding dental applications, various studies have demonstrated the antibacterial activity of CO against *Streptococcus* spp., *S. mutans* [21–23], and other oral pathogens [4]. Likewise, it has been recognized that CO has anti-inflammatory and healing activity in the oral cavity [6] and for acting as a dentin biomodifier [19]. Finally, some studies in animal models report that CO oleoresin is a safe and effective alternative therapy for inflammation and tissue repair in oral wounds [24].

On the other hand, biopolymer-based delivery systems (e.g., nano-composites, microcapsules, emulsions, hydrogels, films, or membranes, among others) have been widely used for the incorporation of bioactive compounds [6,9,20,25–28]. Additionally, membranes or biodegradable films have been designed as controlled-release systems (e.g., would dressing) for various applications in the medical area [6,20,27,29,30]. A biodegradable film is a multi-component system made from polymers (e.g., polysaccharides, lipids, and proteins), which contain in their polymer matrix a plasticizing agent (e.g., glycerol, polyethylene glycol, and sorbitol, among others) and some active compounds of interest. The most widely used polymers for the development of these biomaterials are polysaccharides, such as cellulose and its derivatives, chitosan, pectin, alginate, and recently the use of plant mucilages, among others [31].

Linseed mucilage (*Linum usitatissimum*) is a natural polysaccharide composed of an acidic fraction of pectic-like material (L-galactose, L-fucose, L-rhamnose, and D-galacturonic acid) and a neutral fraction (arabinoxylan, D-galactose, L-arabinose, and D-xylose) [32]. This polymer has been used to produce biodegradable films in various applications, either as an individual polymer [33] or in combination with other polymers, such as chitosan [34,35], pectin [36,37], and polyvinyl alcohol [38]. Likewise, this mucilage has been used to produce active biodegradable films incorporated with carvacrol [39] and *Hamamelis virginiana* extract [31].

Finally, various studies have focused on the production of membranes or active films based on chitosan [20,27,30,40], poly (L-co-D, L lactic acid), and poly (lactic acid)/poly (vinyl pyrrolidone) [6] incorporated with CO for applications in the medical area, mainly as wound dressing. Furthermore, films based on mung bean protein-apple pectin [41] and soy protein isolate loaded with CA [42] have been developed, mainly for applications in the food area.

However, as far as we know, there are no reports of linseed mucilage-based films incorporated with CO and CA for dental applications. Therefore, in this context, this research aimed to produce and characterize films based on linseed mucilage (M) loaded

with *E. cardamomun* (MCA), *C. officinalis* (MCO), and co-loaded with both compounds (MCACO) for future application in the dentistry area.

## 2. Materials and Methods

### 2.1. Vegetal Material

Linseed was purchased at a local supermarket (Mty NL, Mexico). On the other hand, the seeds of *E. cardamomum* (SKU: 209740-01) were purchased from Starwest Botanicals (Sacramento, CA, USA). Finally, *C. officinalis* essential oil was purchased from Young Living Essential Oils, LC Company (Lehi, UT, USA).

### 2.2. Obtaining the Extract of E. cardamomum

The extract was obtained by cold maceration without stirring [43]. Vegetable material (100 g) was placed in a flask containing 400 mL of absolute ethanol (99.5%, CTR Scientific, Mty NL, Mexico; extraction solvent). The sample was left to rest for a period of 24 h, filtered (Whatman™ qualitative filter paper, grade 1), and then placed in a rotary evaporator (IKA RV10, Hayward, CA, USA). Finally, the yield of the extract (% Yield = [final weight of dry extract/initial weight of the plant] × 100) was determined, and the product was stored in a dark container under refrigerated conditions (4.0 ± 2.0 °C) until its later use.

### 2.3. Linseed Mucilage Extraction

The linseed mucilage was extracted according to the methodology reported by Treviño-Garza et al. [31]. Briefly, the flaxseeds (300 g) were placed in distilled water (1000 mL) and kept under constant agitation (250 rpm, 25 ± 2 °C for 2 h). Next, the linseeds were removed with a strainer. Subsequently, ethanol (96%, 2000 mL; CTR Scientific, Mty N.L., Mexico) was added to the resulting aqueous suspension for mucilage precipitation. Finally, the precipitated mucilage was recovered with the help of a strainer and subsequently dried (55 ± 2 °C for 24 h), pulverized, and stored until later use.

### 2.4. Phytochemical Tests of Plant Material

The phytochemical profile of plant material was carried out according to the methodology reported by Guillén-Meléndez et al. [44] and Rodríguez-Garza et al. [45]. In these determinations, the presence (+) or absence (−) of compound groups was analyzed using the following tests: Lieberman–Buchard (sterols, triterpenes), Shinoda (flavonoids), Baljet (sesquiterpene-lactones), sulfuric acid (quinones), ferric chloride (tannins), potassium permanganate (unsaturations), 2,4-dinitrophenylhydrazine (carbonyl group), Dragendorff (alkaloids), sodium hydroxide (coumarins, lactones), Molish (carbohydrates), and foam test (saponins).

### 2.5. Production and Preliminary Characterization of Films Based on Linseed Mucilage Loaded with CA and CO

Four film-forming solutions based on linseed mucilage and loaded with *E. cardamom* (MCA), *C. officinalis* (MCO), and co-loaded with both compounds (MCACO) were designed. Glycerol (99.50% purity, CTR Scientific, Mty N.L., Mexico) was used as a plasticizer and tween 20 (polyoxyethylene-20-sorbitan monolaurate; Sigma Aldrich, St. Louis, MO, USA) as a surfactant agent. The formulations were prepared with the concentrations indicated in Table 1 in distilled water and by constant mechanical stirring (500 rpm, 25 ± 2 °C) until complete dissolution of all the components was achieved. Subsequently, the films were prepared by the casting method; the film-forming solutions (~10 mL) were placed in plastic boxes (60 mm × 15 mm) to be later subjected to a drying process (55 ± 2 °C for 24 h). Finally, the films were manually retrieved and characterized, as indicated in the following sections.

**Table 1.** Film-forming solutions based on linseed mucilage (M) and incorporated with *E. cardamomum* (CA) and *C. officinalis* (CO).

| Formulation | Linseed Mucilage (%) | Glycerol (%) | Tween 20 (%) | *E. cardamomum* (%) | *C. officinalis* (%) |
|---|---|---|---|---|---|
| M | 3.0 | 0.5 | 1.0 | - | - |
| MCA | 3.0 | 0.5 | 1.0 | 2.0 | 0.0 |
| MCO | 3.0 | 0.5 | 1.0 | - | 2.0 |
| MCACO | 3.0 | 0.5 | 1.0 | 1.0 | 1.0 |

### 2.5.1. Color Determination

The color analysis of the films was performed using a colorimeter (Hunterlab model, Colorflex® EZ, Reston, VA, USA), based on the CIE L*a*b* color space (CIELAB; L* coordinate or lightness, 0 black—100 white; coordinate a*, (−) green and (+) red; coordinate b*, (−) blue and (+) yellow) [27,42].

### 2.5.2. Thickness Measurement

Thickness determinations were carried out using a digital micrometer (Quickmike Model Mitutoyo, Kawasaki, Japan); measurements were made at three random positions on each film, and the results were expressed in millimeters (mm) [40,42].

### 2.5.3. Solubility Tests in Artificial Saliva

For the solubility analysis, the films were cut into discs (20 mm in diameter) and placed in beakers containing 50 mL of artificial saliva (Viarden brand, $25 \pm 2$ °C). Subsequently, the solubilization time of the complete films (minutes, min) was determined [31].

### 2.5.4. Antioxidant Activity (DPPH, 1,1-Diphenyl-2-picrylhydrazyl, and ABTS, 2,2′-Azino-bis-3-ethylbenzothiazoline-6-sulfonic Acid) and Total Phenolic Content

The antioxidant activity was calculated by using the DPPH and ABTS free radical (Sigma Aldrich, St. Louis, MO, USA) methods based on previous studies with some modifications [31,46,47]. For sample preparation, films (1 cm × 1 cm) were cut, weighed, and placed in 1 mL of distilled water and vortexed until complete solubilization. Later, the samples were centrifuged (Spectrafuge 6C Labnet International, Inc., Edison, NJ, USA) at 6500 rpm for 10 min to recover the supernatant.

For the DPPH method, the sample (0.75 mL) was placed in conical tubes containing 2.25 mL of the DPPH ethanolic solution (0.039 mg/mL, absorbance = $1.0 \pm 0.005$). The samples were incubated at 25 °C in the dark for a period of 90 min to allow for the reaction. Subsequently, the samples were centrifuged (Spectrafuge 6C, Labnet International, Inc., 6500 rpm for 5–10 min), and the absorbance was determined with a UV-VIS spectrophotometer (Genesys 5, Thermo Spectronic, Rochester, NY, USA) at 517 nm. A calibration curve was made (y = −0.0052x + 0.7161, $R^2$ = 0.99) using Trolox (Sigma Aldrich, St. Louis, MO, USA) as a standard.

For the ABTS method, the sample (0.30 mL) was added to conical tubes containing 2.70 mL of the ABTS radical solution (ABTS, 7.00 mM, and potassium persulfate, 2.45 mM, 1:1 ratio, absorbance = $0.7 \pm 0.005$). The samples were incubated at 25 °C in the dark for a period of 7 min to allow for the reaction. Subsequently, the samples were centrifuged (Spectrafuge 6C, Labnet International, Inc., 6500 rpm for 5–10 min), and the absorbance was determined with a UV-Vis spectrophotometer (Genesys 5, Thermo Spectronic, Rochester, NY, USA) at 734 nm (ABTS). A calibration curve was made (y = −0.0024x + 0.577, $R^2$ = 0.99) using Trolox (Sigma Aldrich, St. Louis, MO, USA) as a standard.

Finally, the results were expressed as µM Trolox equivalents (TE)/g of film for both methods. Additionally, the antioxidant activity was also determined by the percentage of inhibition, according to the following equation: inhibition scavenging activity (%) =

(absorbance of the radical solution − absorbance of the sample/absorbance of the radical solution) × 100.

The total phenolic content of the films was calculated using the Folin–Ciocalteu (Sigma Aldrich, St. Louis, MO, USA) technique with some modifications [48,49]. The samples (1.6 mL) were placed in conical polypropylene tubes (15 mL). Subsequently, the Folin–Ciocalteu reagent (0.1 mL) and sodium carbonate (20% $w/v$, 0.3 mL, CTR Scientific, Mty N.L., Mexico) were added, and the solutions were homogenized in a vortex (Mixer Labnet International, Inc., Edison, NJ, USA). The samples were kept at rest in dark conditions (90 min at 25 °C) and then were centrifuged (Spectrafuge 6C Labnet International, Inc., Edison, NJ, USA; 6500 rpm for 10 min) and analyzed at 760 nm in a UV-Vis spectrophotometer (Genesys 5, Thermo Spectronic, Rochester, NY, USA). Finally, a calibration curve was made using gallic acid (Sigma Aldrich, St. Louis, MO, USA) as a standard (y = 0.07x − 0.2533 $R^2$ = 0.9958), and the total phenolic content was expressed as µg of gallic acid equivalents (GAE)/g of film.

### 2.5.5. Texture Profile Analysis (APT)

The texture analysis of the films was carried out with the help of a texturometer (Brookfield, CT3, Middleboro, MA, USA). The films (60 mm in diameter) were placed in the equipment and analyzed with the TA44 probe (cylinder 4 mm diameter), with an activation load of 0.070 N, a test speed of 0.50 mm/s, and a load range of 1000 g. The parameters evaluated were hardness (N; newtons), percentage deformation (%), adhesiveness (mJ; millijoules), and fracturability (N; newtons).

### 2.5.6. Field Emission Scanning Electron Microscopy (FE-SEM)

Film microscopy (10 × 10 mm) was performed with a field emission scanning electron microscope (Zeiss Sigma 300 VP, Jena, Germany) under high vacuum conditions, with a secondary electron detector considering a voltage of 5 kV, and at a working distance of 6 mm. Prior to analysis, the films were coated with gold (sputtering) to obtain better imaging conditions. Finally, micrographs were taken at 200× and 1500× magnification.

### 2.6. Statistical Analysis

The results of color, thickness, solubility, antioxidant activity, total phenols, and texture profile were analyzed by analysis of variance (ANOVA) and Tukey's test, with a significance level of $p \leq 0.05$, using the SPSS software (IBM version 22, SPSS Inc., Chicago, IL, USA).

## 3. Results and Discussion

### 3.1. Phytochemical Analysis of Plant Extracts

In the first place, the yield of the CA extract was 5.80%, in agreement with that reported by Cárdenas-Garza et al. [2]. Moreover, the phytochemical analysis showed that the plant material contains a complex mixture of various components [45]. Table 2 shows that *E. cardamomum* and linseed mucilage were positive for eight of the phytochemical tests, except for quinones, coumarins, and saponins. Additionally, the chemical composition of the cardamom extract (α-terpinyl acetate, mainly) has been reported in previous studies by our working group [2]. In addition, *C. officinalis* was positive only for six tests performed (sterols and triterpenes, flavonoids, unsaturations, carbonyl group, alkaloids, and carbohydrates). These results agree with what was reported by Hanaa et al. [50] and Yasmeen et al. [51], who have found some of these components in the linseed. In addition, the presence of some of these phytochemical compounds has also been reported in the essential oil of *C. officinalis* [52] and natural extracts of *E. cardamomum* [2,53,54]. Finally, this type of phytochemical compound has been characterized by having multiple pharmacological properties, among which stand out anti-inflammatory, anticancer, antiseptic, antimicrobial, and antioxidant activities, among others [2,4]. Likewise, due to its composition, copaiba oil has been characterized by reducing the formation of dental biofilm [21], thus counteracting some oral pathogens [4], presenting an anti-inflammatory effect and healing in the oral

cavity [55], and acting as a dentin biomodifier [19]. On the other hand, cardamom is efficient in reducing the levels of microbial viability in the dental biofilm [15], having a potential therapeutic effect on periodontal infections [17], and improving oral hygiene [16].

**Table 2.** Partial phytochemical screening of plant extracts.

| Tests | Phytoconstituents | Reaction | CA | CO | M |
|---|---|---|---|---|---|
| Liebermann-Burchard | Sterols, triterpenes | Reddish-brown ring | + | + | + |
| Shinoda | Flavonoids | Reep red | + | + | + |
| Baljet | Sesquiterpene-Lactones | Orange color | + | − | + |
| Sulfuric acid ($H_2SO_4$) | Quinones | Red color | − | − | − |
| Ferric chloride ($FeCl_3$) | Phenols, tannins | Green color | + | − | + |
| Potassium permanganate ($KMnO_4$) | Unsaturations | Brown precipitate | + | + | + |
| DNPH (2,4-dinitrophenylhydrazine) | Carbonyl group | Orange color | + | + | + |
| Dragendorff | Alkaloids | Orange precipitate | + | + | + |
| Sodium hydroxide (NaOH) | Coumarins, lactones | Yellow color | − | − | − |
| Molish | Carbohydrates | Purple ring formation | + | + | + |
| Foam | Saponins | Presence of stable foam | − | − | − |

*3.2. Development and Partial Characterization of Films Based on Linseed Mucilage Loaded with CA and CO*

3.2.1. Film Production

The control films and those incorporated with the extracts (MCA, MCO, MCACO) were successfully produced (Table 3). The films obtained (60 mm in diameter) were thin and slightly flexible, with an opaque appearance, a light brown hue (control and MCO), and a more intense brown color (MCA and MCOCA). These results are similar to those reported in previous studies [30,31,42].

**Table 3.** Macroscopic characteristics of films based on linseed mucilage (M) and incorporated with *E. cardamomum* (CA) and *C. officinalis* (CO).

| Formulation | Representative Image | Homogeneity | Appearance |
|---|---|---|---|
| MCA |  | Intermediate | Opaque, dark brown, and slightly flexible |
| MCO |  | Intermediate | Opaque, slightly brown, and slightly flexible |
| MCACO |  | Low | Opaque, light brown, and slightly flexible |
| Control |  | Intermediate | Opaque, slightly brown, and slightly flexible |

3.2.2. Physicochemical Properties

Color and Thickness

The color properties affect the visual appearance and could influence the patient's perception of the product. Regarding color, no significant difference ($p > 0.05$) was found in

the luminosity values between the different treatments, whose values fluctuated between 25.33 and 32.93 (Figure 1). This finding agrees with the literature since Hajirostamloo et al. [42] reported that the L* values remain constant at a concentration of 1% CO but increase at a concentration of 5% in protein isolate films containing cardamom essential oil. Likewise, Rodrigues et al. [56] indicated that the addition of CO oil (10%) does not affect the luminosity of starch-based films incorporated with copaiba oil. On the contrary, for the values of a* and b*, a significant difference ($p < 0.05$) was found between the treatments; the values in both coordinates were highest for MCA (a* = 1.60 ± 0.98 and b* = 5.93 ± 1.25) and MCOCA (a* = 1.14 ± 0.19 and b* = 6.27 ± 0.76), while the lowest values were for MCO (a* = −0.40 ± 0.03 and b* = 1.05 ± 0.88) and the control (a* = −0.31 ± 0.02 and b* = 1.31 ± 0.68) (Figure 1). According to what was reported by Hajirostamloo et al. [42], the incorporation of CO into the films affected the optical attributes, increasing the values of a* and b* coordinates, as observed in the MCO and MCACO treatments. On the other hand, Rodrigues et al. [56] reported that the color of the films was directly related to the color of the polymer and the oil or extract incorporated into the formulation. In addition, they also reported that the CO oil provided a slightly yellowish coloration to the films, which was more accentuated as the concentration of CO in the polymer matrix increased. Therefore, it is likely that, since the color of CO is similar to that of linseed mucilage, the MCO and control films are very similar in the a* and b* coordinate values.

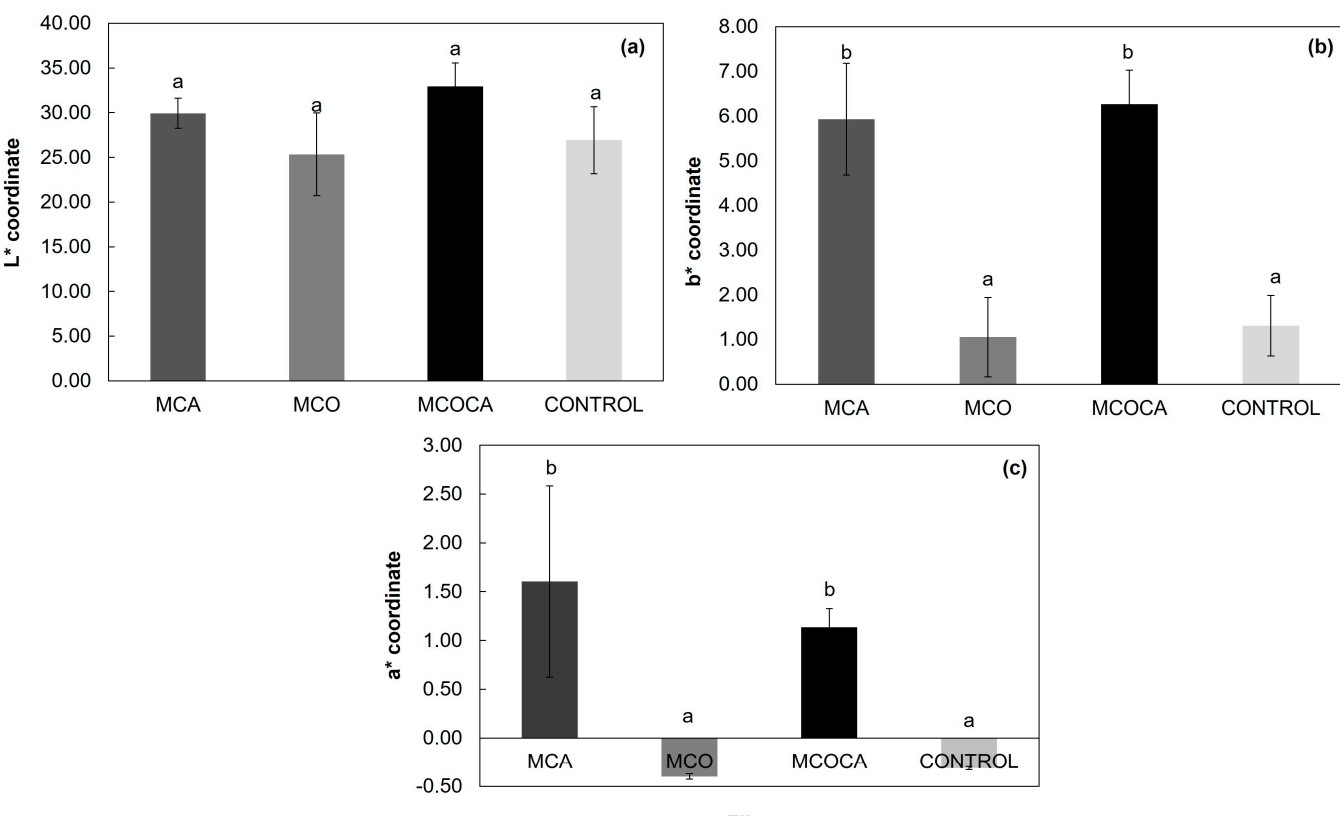

**Figure 1.** Color properties (**a**) luminosity, (**b**) blueness (−)—yellowness (+), and (**c**) redness (−)—greenness (+) of films based on linseed mucilage (M) and incorporated with *E. cardamomum* (CA) and *C. officinalis* (CO). Different letters (a,b) indicate significant differences ($p < 0.05$) between treatments.

The thickness is an important parameter since thin films tend to be less uncomfortable for the patient. In the thickness parameter, no significant difference ($p < 0.05$) was found between the treatments, and the values fluctuated between 0.0043 and 0.0052 mm (Figure 2). These results agree with those found by Herrera Brandelero et al. [57] and Rodrigues

et al. [56], who reported that the incorporation of CO oil did not modify the thickness of films made from starch/PVOH/alginate and starch, respectively. Moreover, Hajirostamloo et al. [42] reported an increase in the thickness of the films when incorporating CA; this parameter increases with the increase in the amount of extract (5–20%) in a polymeric matrix of soy protein isolate.

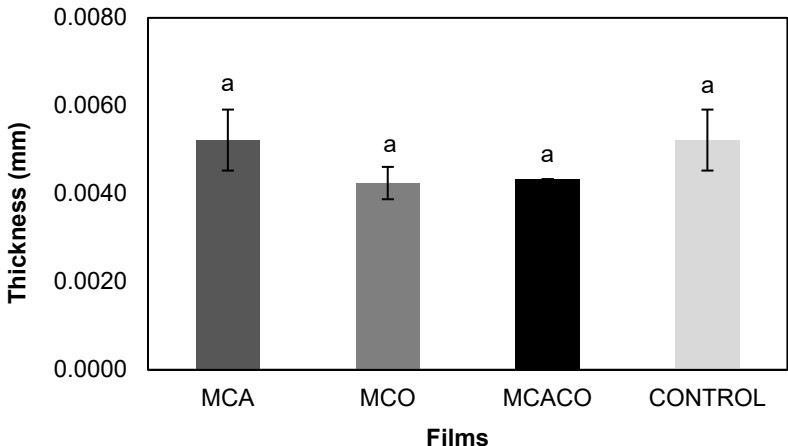

**Figure 2.** Thickness of films based on linseed mucilage (M) and incorporated with *E. cardamomum* (CA) and *C. officinalis* (CO). Letters (a) indicate no significant differences ($p > 0.05$) between treatments.

Solubility Tests in Artificial Saliva

In the case of the solubility of the films, although the values were higher for MCO ($12.00 \pm 2.65$ min) and MCA ($9.00 \pm 1.73$ min), no significant difference ($p < 0.05$) was found between the treatments (MCOCA = $12.33 \pm 1.15$ and control = $8.67 \pm 0.58$), indicating that the incorporation of the extracts into the polymeric matrix does not modify this property (Figure 3).

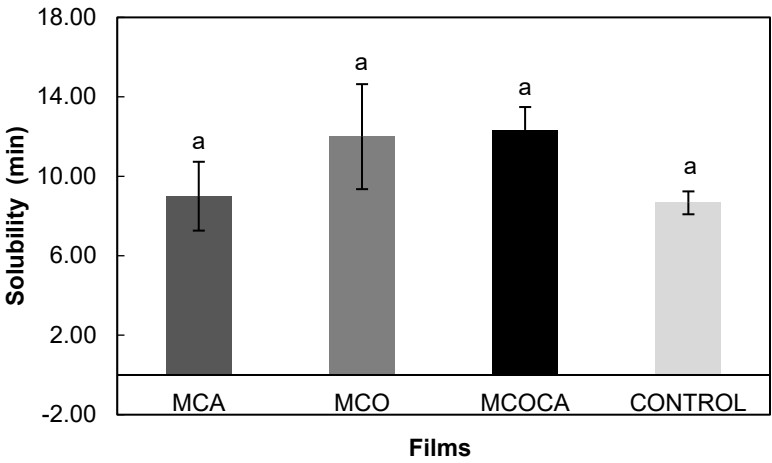

**Figure 3.** Solubility in artificial saliva of films based on linseed mucilage (M) and incorporated with *E. cardamomum* (CA) and *C. officinalis* (CO). Letters (a) indicate no significant differences ($p > 0.05$) between treatments.

According to what was reported by Puligundla and Lim [32], linseed mucilage shows a high degree of swelling and solubility in aqueous solutions. The water absorption leads to swelling and solubilization of the films, allowing the complete release of the active compounds incorporated within the polymeric matrix towards the artificial saliva. This behavior is directly associated with the hydrophilic nature of the polysaccharide and with the slightly branched structure of the linseed mucilage, which leads to the film dissolving and losing its structure over time [37]. In addition, the difference in the solubility time of

the MCO and MCACO films could be related to the low solubility of the CO oil in water (the main component of artificial saliva) [3,26,30]. Likewise, Rodrigues et al. [56] reported that the addition of CO oil decreased the hydrophilic character of the films, leading to lower solubility. This behavior can be attributed to the fact that the interaction between the hydroxyl groups of the polymeric matrix and the oil components makes the -OH groups less available and, consequently, the solubility time of the films in artificial saliva increases.

Antioxidant Activity (DPPH and ABTS) and Total Phenolic Content

Regarding the antioxidant activity of the films, significantly higher values ($p < 0.05$) were for MCA (DPPH = 5781.38 ± 92.10 µm TE/g, 81.99 ± 3.56% inhibition and ABTS = 11,855.82 ± 1159.46 µm TE/g, 95.80 ± 3.98% inhibition), followed by MCACO treatment (DPPH = 4345.29 ± 150.87 µm TE/g, 47.15 ± 1.46% inhibition and ABTS = 5598.14 ± 1051.41, 39.73 ± 7.77% inhibition). The lowest values were for MCO (DPPH = 1740.18 ± 236.51 µm TE/g, 21.78 ± 4.83% inhibition and ABTS = 5174.77 ± 1487.98 µm TE/g, 35.38 ± 6.22% inhibition) and the control (DPPH = 2132. 39 ± 229.87 µm TE/g, 23.85 ± 2.57% inhibition and ABTS = 4615.74 ± 154.00 µm TE/g, 34.04 ± 1.66% inhibition) (Figure 4).

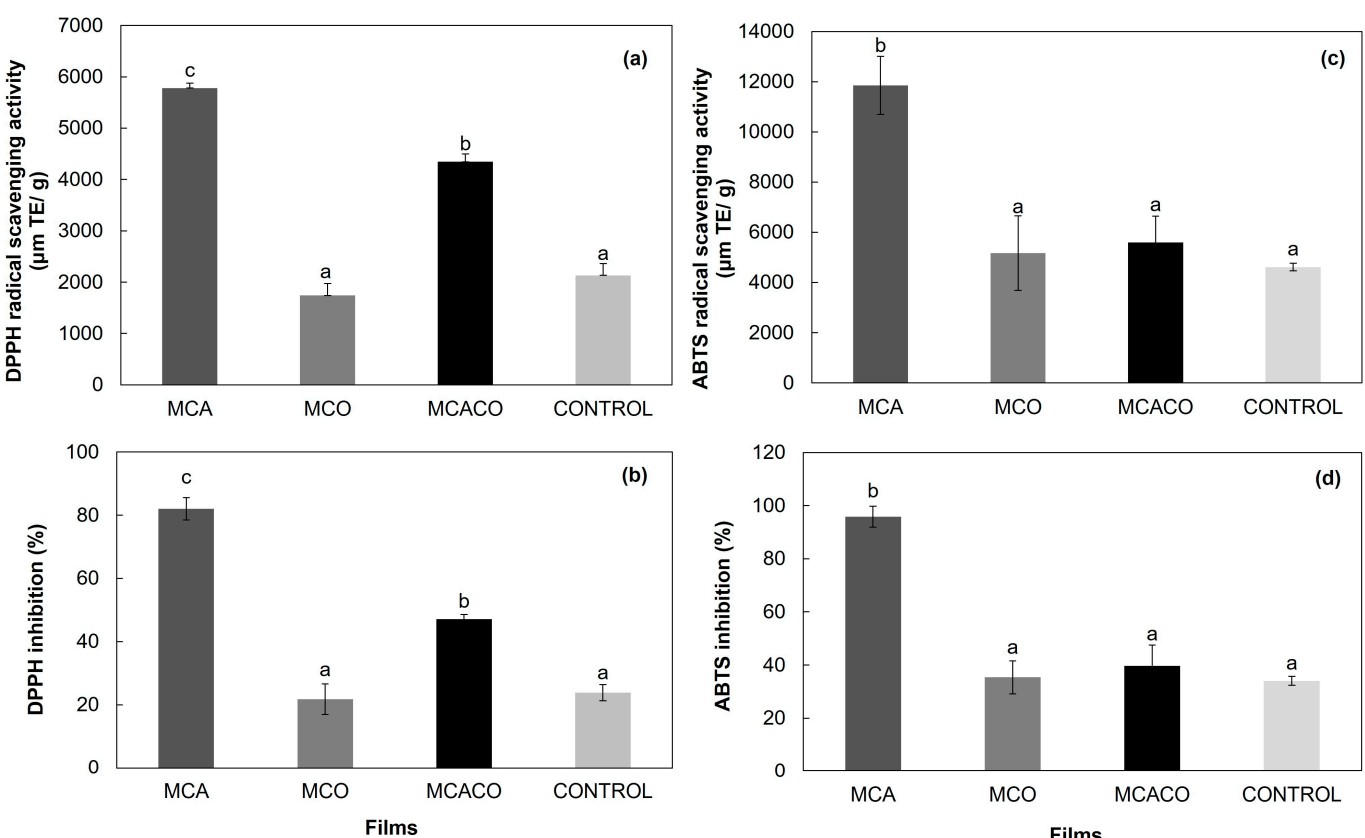

**Figure 4.** (**a**,**b**) DPPH and (**c**,**d**) ABTS radical scavenging activity and inhibition (%) of films based on linseed mucilage (M) and incorporated with *E. cardamomum* (CA) and *C. officinalis* (CO). Different letters (a,b,c) indicate significant differences ($p < 0.05$) between treatments.

These results agree with what was found by Hajirostamloo et al. [42], who reported that the incorporation of CA significantly increased the antioxidant activity of the films, which is accentuated by the increase in the concentration of CA in the polymeric matrix (% of inhibition of 9.10–63.61% with concentrations of 1–20% of CA, respectively), as we found in the MCA and MCACO films (Figure 4). Furthermore, the antioxidant activity provided by CO has also been previously reported in other applications [58], similar to that found in MCO films. Finally, the antioxidant activity reported in the control films is higher than that

reported in other films made from linseed mucilage [31]; this behavior could be related to the polymer extraction processes, as well as the variety of seeds used for its recovery.

In the case of the total phenolic content (Figure 5), the highest values were for MCA (8076.49 ± 297.66 µg GAE/g), followed by MCACO (3985.25 ± 213.21 µg GAE/g), while the lowest values were for MCO and the control (3270.18 ± 243.99 and 2831.16 ± 392.69 µg GAE/g, respectively). These results agree with what was found in the DPPH and ABTS antioxidant activity analyses (Figure 4), which reflect a higher activity in MCA and MCACO, which can be associated with a higher total phenolic content in CA. In general, the antioxidant properties of the films developed in this study can be attributed to the content of antioxidant compounds present in the extract, oil, and linseed mucilage, such as sterols, triterpenes, flavonoids, tannins, phenols, and alkaloids, among others (Table 1), which have been reported in previous studies [2,18,52,59,60]. In addition, the mechanism for the action of these compounds is associated with the donation of a hydrogen atom and/or electrons to free radical chemical species to prevent cell damage by reducing oxidative stress, reactive oxygen species (ROS) and nitrogen (RON), oxidation of lipids, proteins, and DNA [58,61].

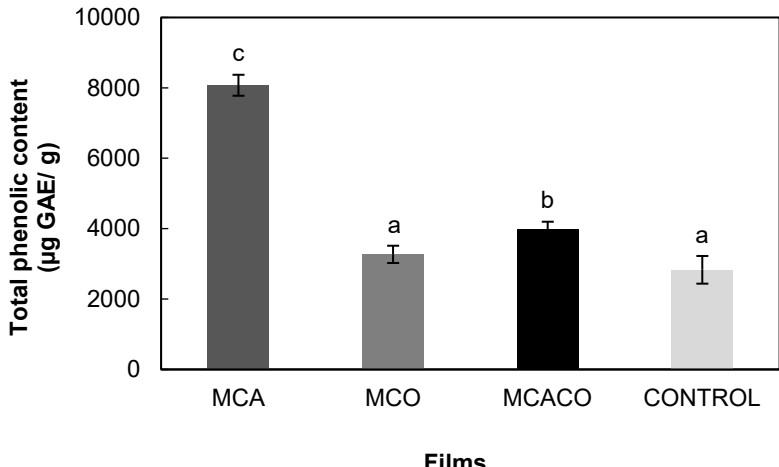

**Figure 5.** Total phenolic content of films based on linseed mucilage (M) and incorporated with *E. cardamomum* (CA) and *C. officinalis* (CO). Different letters (a,b,c) indicate significant differences ($p < 0.05$) between treatments.

*3.3. Texture Profile Analysis (TPA)*

Hardness, Deformation, Adhesiveness and Fracturability

The textural properties of the films provide information to understand the behavior of materials. Regarding the texture analysis of the films, it was observed that the incorporation of CA or CO decreased ($p < 0.05$) the hardness of the films (Figure 6a). Hardness refers to the maximum force required to compress the films between the molars, the tongue, or the palate [62]. The control film presented the highest hardness value (8.86 ± 1.54 N), and the MCA film had the lowest value (2.59 ± 0.35 N). In addition, the MCOCA and MCO films presented similar values, fluctuating between 4.78 and 5.35 N. According to the report by [31], the incorporation of extracts and essential oils into the films provides them a plasticizing effect since they reduce the interchain interaction forces of the polymer, increasing the mobility of the chains and providing greater flexibility. Consequently, the film has a less rigid and more malleable structure; this behavior can be attributed to the presence of hydroxyl groups (-OH) that form H bonds with the mucilage, increasing the mobility and decreasing the hardness of the polymeric matrix.

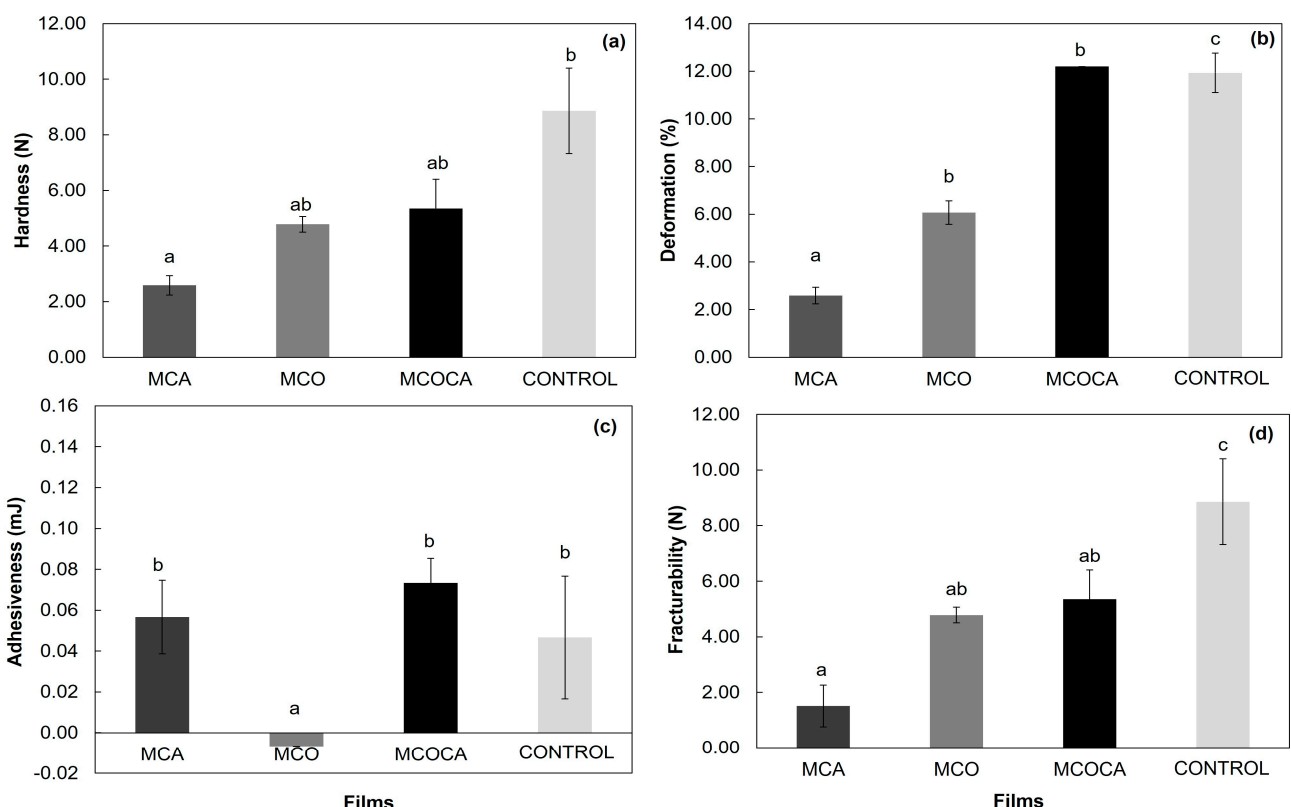

**Figure 6.** Textural properties: (**a**) hardness, (**b**) deformation, (**c**) adhesiveness and (**d**) fracturability of films based on linseed mucilage (M) and incorporated with *E. cardamomum* (CA) and *C. officinalis* (CO). Different letters (a,b,c) indicate significant differences ($p < 0.05$) between treatments.

Furthermore, it was found that the incorporation of CA and CO significantly decreased ($p < 0.05$) the deformation of the films (Figure 6b). In general, the deformation of the film occurs when the applied compressive stress changes the internal structure of the material, which prevents it from returning to its structural or dimensional original. The highest deformation values were found in MCACO and control treatments (11.93 ± 0.83% and 12.20 ± 0.00%, respectively). The lowest deformation values were obtained for MCA and MCO (2.59 ± 0.35% and 6.07 ± 0.49%, respectively). The improvement in this property could be related to the ability of the essential oil or extract to induce the reorganization of the polymeric network in the film matrix, which leads to the formation of films with a more stable structure [63].

The adhesiveness parameter refers to the work required to overcome the attractive force between the film and a surface [62]. The lowest value ($p < 0.05$) was obtained for the MCO treatment (−0.01 ± 0.00 mJ). The control, MCA, and MCO films presented similar values that fluctuated between 0.05 and 0.07 mJ (Figure 6c). In general, most of the films (MCA, MCOCA, and control) showed very similar adhesiveness values; the low values found for MCO could be related to the characteristics of the CO oil, which influence the final properties of the film.

Finally, fracturability refers to the force necessary to fracture the film [62]. In this parameter, it was found that the incorporation of the extracts decreased the fracturability of the films. The highest values were for the control (8.86 ± 1.54 N), followed by MCOCA (5.35 ± 1.05 N), MCO (4.78 ± 0.28 N), and MCA (1.51 ± 0.75) (Figure 6d). As previously mentioned, the incorporation of essential oils and extracts into the film matrix can cause a rearrangement of the polymer chains, as well as a plasticizing effect that provides a film with a more stable and fracture-resistant polymer structure [63].

### 3.4. Microscopic Analysis

The FE-SEM analysis is an essential tool for analyzing the films at the microscopic level. Regarding the microscopic analysis of the control and MCO, the films presented a more homogeneous microstructure, as well as a smooth (with few bulges) and continuous surface, similar to that reported by Karami et al. (chitosan-linseed mucilage films) [35], Debone et al. (chitosan/copaiba oleoresin films) [27], Norcino et al. (pectin films loaded with copaiba oil nanoemulsions) [36], and Pinto et al. (copaiba essential oil loaded-nanocapsules films) [40]. Otherwise, the MCA and MCACO films presented a more heterogeneous microstructure, a rough surface with the presence of discontinuous matter (bulges) in the polymer matrix associated with the presence of components of the CA extract; this effect has also been reported in protein isolate films containing cardamom essential oil [42] (Figure 7).

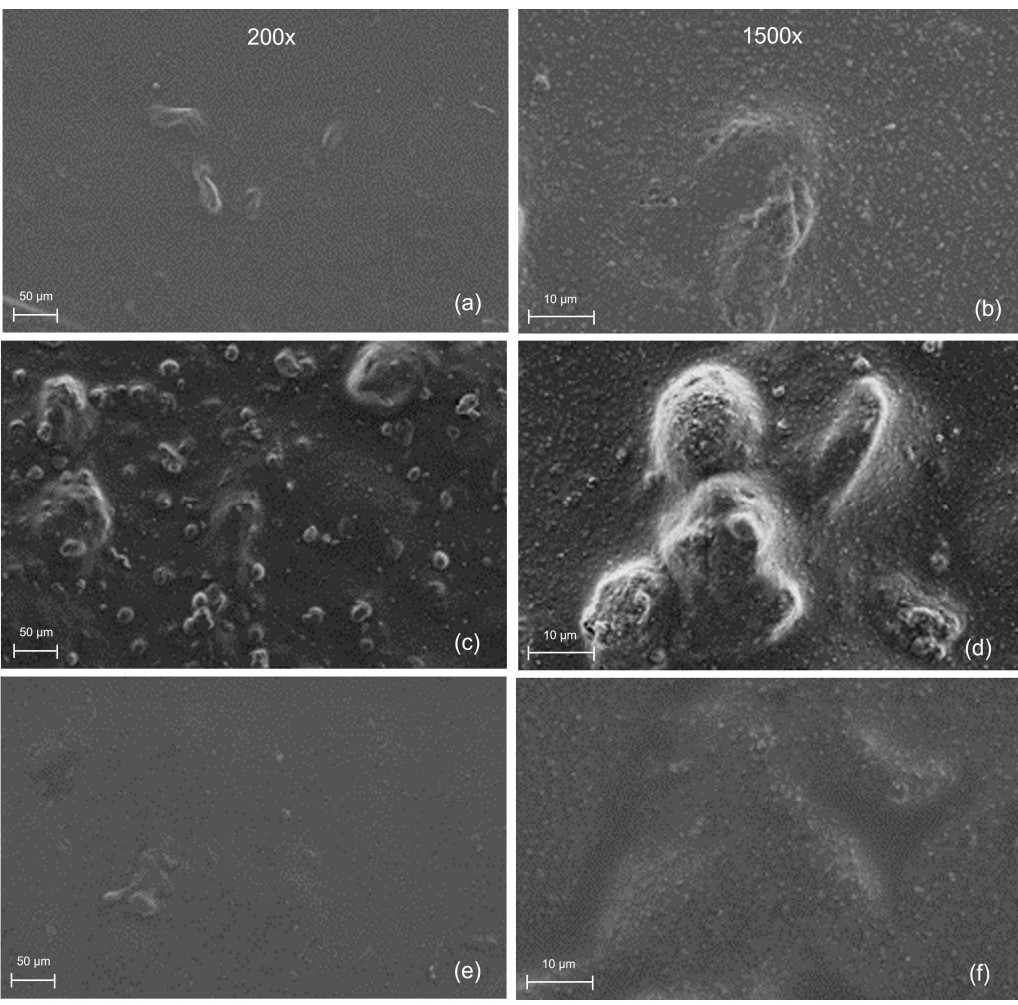

**Figure 7.** *Cont.*

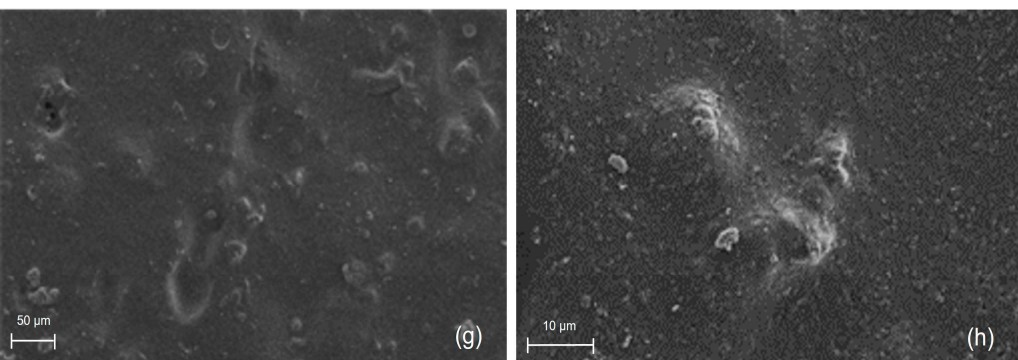

**Figure 7.** Surface micrographs (200× and 1500×) of films based on linseed mucilage (M) and incorporated with *E. cardamomum* (CA) and *C. officinalis* (CO); control (**a**,**b**), MCA (**c**,**d**), MCO (**e**,**f**), and MCOCA (**g**,**h**).

## 4. Conclusions

In their composition, *E. cardamomum* and *C. officinalis* compounds had evidence of the presence of phytochemicals, such as sterols, triterpenes, flavonoids, alkaloids, tannins, and phenols, among others. Moreover, it was possible to produce linseed mucilage-based films incorporated with *E. cardamomun*, *C. officinalis*, and co-loaded with both bioactive compounds. The addition of the active compounds in the films modified the color (redness–greenness and yellowness). However, the thickness parameter remained constant in all treatments (0.0042–0.0052 mm). In addition, the solubility time was higher for films containing CO (~12 min). Furthermore, the addition of the bioactive compounds increased the antioxidant activity and the total phenolic content, mainly in the films incorporated with CA. Regarding the texture analysis, the incorporation of the active ingredients decreased the hardness (39.50%–70.81%), deformation (49.16%–78.30%), and fracturability (39.58%–82.95%); however, it did not modify the adhesiveness, except in MCO. On the other hand, the FE-SEM micrographs showed a more homogeneous structure for the MCO films with respect to the films that contained CA in the formulation (heterogeneous structure with the presence of protuberances). Finally, due to the previously reported pharmacological properties of *E. cardamomun* and *C. officinalis* (anti-inflammatory, antimicrobial, antioxidant, among others), the films developed in this research could have a potential application in the area of dentistry as a wound dressing. However, it is necessary to continue with this research to evaluate the mechanical (tensile strength and elongation at break), antimicrobial, anti-inflammatory, and cytotoxic properties, as well as consider the development of in-vivo studies to broadly understand the properties of the films.

**Author Contributions:** Conceptualization, O.E.R.-L., M.Z.T.-G. and A.K.S.-V.; methodology, A.K.S.-V., M.Z.T.-G., J.G.B.-G., J.H.E.-L. and M.d.R.L.-B.; software, O.E.R.-L., J.G.B.-G. and M.Z.T.-G.; validation, O.E.R.-L., J.G.B.-G. and M.Z.T.-G.; formal analysis, O.E.R.-L., M.Z.T.-G., A.C.-M., S.M.L.-V., J.G.B.-G., J.H.E.-L. and M.d.R.L.-B.; investigation, A.K.S.-V., M.Z.T.-G., J.G.B.-G. and O.E.R.-L.; resources, O.E.R.-L., J.G.B.-G. and M.Z.T.-G.; data curation, O.E.R.-L., J.G.B.-G. and M.Z.T.-G.; writing—original draft preparation, M.Z.T.-G. and A.K.S.-V.; writing—review and editing, M.Z.T.-G., O.E.R.-L. and J.G.B.-G.; visualization, O.E.R.-L., A.K.S.-V. and M.Z.T.-G.; supervision, M.Z.T.-G., O.E.R.-L. and J.G.B.-G.; project administration, O.E.R.-L. and M.Z.T.-G.; funding acquisition, O.E.R.-L., M.Z.T.-G. and J.G.B.-G. All authors have read and agreed to the published version of the manuscript.

**Funding:** This research was funded by the Scientific and Technological Research Support Program of the Universidad Autónoma de Nuevo Leon (UANL), PAICYT CT1600-21 and CN1930-21.

**Institutional Review Board Statement:** Not applicable.

**Informed Consent Statement:** Not applicable.

**Data Availability Statement:** Not applicable.

**Acknowledgments:** To the CONAHCYT, National Council of Humanities, Sciences, and Technologies, for the support granted through scholarship 003989.

**Conflicts of Interest:** The authors declare no conflict of interest.

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
