# Peer review of "Production and Preliminary Characterization of Linseed Mucilage-Based Films Loaded with Cardamom (Elettaria cardamomum) and Copaiba (Copaifera officinalis)"

_coatings, doi:10.3390/coatings13091574_

Round 1
Reviewer 1 Report
Article may be accepted after addressing the following comments:
1. Abstract part needs to rewrite add major numerical data.
2. In the keywords add major content/properties studied
3. Add manufacturer of chemicals used in this studies.
4. Add floe diagram for extraction of linseed mucilage
5. Write the method of antioxidant and total phenolic content in brief.
6. Reanalyze the texture properties and determine elongation at break and tensile
Strength.
7. Article is looks like the report, rewrite this article firstly discuss the values observed and then provide the reason/application/benefit behind the results observed.
8. Maximum data is shown in graph form; show some data in tabular form.
9. Language is poor, rewrite carefully
Language is poor, improve it.
Author Response
We appreciate the comments and suggestions of the reviewers regarding the manuscript entitled “Production and preliminary characterization of linseed mucilage-based films loaded with cardamom (Elettaria cardamomum), and copaiba (Copaifera officinalis)”. The suggestions and modifications indicated by the reviewers were made in the manuscript downloaded from the link using "track changes", the responses for the reviewers are shown below. Additionally, we clarify that before submission the manuscript was subjected to an English language review.
Article may be accepted after addressing the following comments:
We appreciate the comments and suggestions of reviewer 1.
- Abstract part needs to rewrite add major numerical data.
Ok. It was modified.
- In the keywords add major content/properties studied
Ok. It was modified.
- Add manufacturer of chemicals used in this studies.
The manufacturers of the chemicals were included throughout the manuscript when naming the different chemical compounds in the materials and methods section.
- Add floe diagram for extraction of linseed mucilage
We appreciate the suggestion. We made the flowchart (view pdf document). However, we consider that the methodology of the mucilage extraction process is short. We would like to keep the extraction method of the linseed mucilage as it is in its current state.
- Write the method of antioxidant and total phenolic content in brief.
Ok. It was modified.
- Reanalyze the texture properties and determine elongation at break and tensile Strength.
We appreciate the suggestions of reviewer 1. We know that the analysis of tensile strength and elongation at break parameters is desirable. However, at this moment, we don´t have the equipment in our lab to perform these analyses. Therefore, we decided to make a texture analysis of the films. We believe that these results can also provide useful information to understand the behavior of materials. We also consider modifying the title and adding the terminus “preliminary characterization” and recommend that future investigations realize the tensile strength and elongation analysis.
- Article is looks like the report, rewrite this article firstly discuss the values observed and then provide the reason/application/benefit behind the results observed.
Ok. We made some modifications to the color, thickness, solubility, texture profile analysis and microscopic analysis sections. In general, we consider most of the sections to mention the importance of the analysis, describe the results, compare them with the literature, and understand the reason for them. We believe it is valid.
- Maximum data is shown in graph form; show some data in tabular form.
We appreciate the suggestion. However, as a matter of style, we would like to keep the results as displayed in their current state.
- Language is poor, rewrite carefully
Ok. It was revised. The article was submitted for an English language review.

Reviewer 2 Report
This paper is based on the preparation of environmentally friendly plant-based chemical films for dental materials. The sufficient and appropriate characterization proves the feasibility of the prepared thin film. Thus, it can be published after dealing with the following issues.
1. The definitions of parameters "a*" and "b*" should be indicated in the Abstract Section. Due to the expressed intention, these should be the color items that describe "yellow/blue" and "red/green" couples (as explained in Section 2.5.1). In addition, the application implied for these films is dentistry purpose; thus, please make more explanation on this, and the last sentence "however, it is necessary to continue this research to learn more about the properties and use of films" can be deleted.
2. The feasibility of this study for medical purpose is insufficiently introduced. For instance, how the mentioned films work in the stomatology? Please supplement more descriptions. As a strong background introduction, the various application of plant extracts should be stated in the initial of context.
3. What is the extraction efficiency of the employed cold maceration strategy? Are the extracted products (CA and CO) stable in the ambient environment? Usually, the expression "vegetable material" is not common; I think "plant extract" is more proper. In addition, if possible, please conduct an explicit phytochemical analysis in Section 3.1 (e.g., HPLC, based on the standard chemicals) to boost the scientific level of this paper. Also, the dimension of images in Table 3 should be marked, by this way, readers can catch the feasibility of the films.
4. For the practical meaning, the basic mechanical performance of as-prepared films should be tested, at least the tensile strength. In addition, is there any realistic effect of these or the optimum film for teeth?
5. The controlled release mechanism of as-prepared films should be discussed in-depth, which is also mentioned in the Introduction Section (from line 72). In the discussion part, authors can refer to the literatures that always have the detailed classification on this.
6. How are the durance of as-developed films? In other words, can the films be maintained long-term antioxidant activity in the presence of hash ambient conditions (e.g., water immersion, CO2 attack, …)
7. The language of this manuscript should be polished thoroughly for trivial grammar and stylistic errors.
Author Response
We appreciate the comments and suggestions of the reviewers regarding the manuscript entitled “Production and preliminary characterization of linseed mucilage-based films loaded with cardamom (Elettaria cardamomum), and copaiba (Copaifera officinalis)”. The suggestions and modifications indicated by the reviewers were made in the manuscript downloaded from the link using "track changes", the responses for the reviewers are shown below. Additionally, we clarify that before submission the manuscript was subjected to an English language review.
This paper is based on the preparation of environmentally friendly plant-based chemical films for dental materials. The sufficient and appropriate characterization proves the feasibility of the prepared thin film. Thus, it can be published after dealing with the following issues.
We appreciate the comments and suggestions of reviewer 2.
- The definitions of parameters "a*" and "b*" should be indicated in the Abstract Section. Due to the expressed intention, these should be the color items that describe "yellow/blue" and "red/green" couples (as explained in Section 2.5.1). In addition, the application implied for these films is dentistry purpose; thus, please make more explanation on this, and the last sentence "however, it is necessary to continue this research to learn more about the properties and use of films" can be deleted.
We agree. It was modified.
- The feasibility of this study for medical purpose is insufficiently introduced. For instance, how the mentioned films work in the stomatology? Please supplement more descriptions. As a strong background introduction, the various application of plant extracts should be stated in the initial of context.
We appreciate the suggestions and comments of the reviewer. Films work as release systems (would dressing). The introduction section includes some studies (lines 94 – 97) where copaiba-based films were developed for medical applications, mainly as would dressing (it was added in the text).
Moreover, plant extracts are indicated for antimicrobial, antifungal, antiseptic, anti-inflammatory, and antioxidant, among other purposes (lines 66 - 69). In dentistry, it has been reported that cardamom possesses anti-bad breath, anticaries, antiseptic, and antimicrobial properties. Multiple studies have shown that cardamom has been effective against microorganisms that cause infections and dental caries, such as Streptococcus mutans, Candida albicans, and Lactobacillus casei. Additionally, it has been shown that the use of cardamom in combination with other extracts (black pepper/ black cumin/ cardamom and black pepper/ black cumin/ cardamom/cinnamon) has a high antibacterial susceptibility against microorganisms from oral isolates. Finally, some reports indicated the potential therapeutic benefits of cardamom for periodontal infections (lines 55 – 63).
Additionally, various studies have demonstrated the antibacterial activity of copaiba against Streptococcus spp., S. mutans, and other oral pathogens. Also, likewise, it has been recognized that copaiba has anti-inflammatory and healing activity in the oral cavity and for acting as a dentin bio modifier. Finally, some studies in animal models report that copaiba oleoresin is a safe and effective alternative therapy for inflammation and tissue repair in oral wounds (lines 67 – 74). The information mentioned above was included in the introduction section.
- What is the extraction efficiency of the employed cold maceration strategy? Are the extracted products (CA and CO) stable in the ambient environment? In general, although the yield of CA extraction is low (5.80%), the efficiency of the cold maceration method is associated with the fact that it is possible to extract most of the properties of the plant without alterations because of temperature and that the number of degraded compounds is minimal. Although the objective of this investigation was not to carry out an evaluation of the useful life of the extracts. In general, we know that the extract can be kept, even for months, if it is kept in dark conditions and in a cool place (~ 25 °C).
Usually, the expression "vegetable material" is not common; I think "plant extract" is more proper. Ok. It was modified.
In addition, if possible, please conduct an explicit phytochemical analysis in Section 3.1 (e.g., HPLC, based on the standard chemicals) to boost the scientific level of this paper. Studies of the chemical composition of cardamom extract by GC-MS/MS have been reported in previous studies by our work group Cárdenas Garza et al., [2]. This information was added in the discussion section. Moreover, soon, we plan to work on the chemical characterization of copaiba. However, we do not consider it for this research.
Also, the dimension of images in Table 3 should be marked, by this way, readers can catch the feasibility of the films. Ok. The dimensions of the films obtained (60 mm) were specified in section 3.2.1. Film production.
- For the practical meaning, the basic mechanical performance of as-prepared films should be tested, at least the tensile strength. In addition, is there any realistic effect of these or the optimum film for teeth?
We appreciate the suggestions of reviewer 2. We know that the analysis of tensile strength is desirable. However, at this moment, we don´t have the equipment in our lab to perform these analyses. Therefore, we decided to make a texture analysis of the films. We believe that these results can also provide useful information to understand the behavior of materials. We also consider modifying the title and adding the terminus “preliminary characterization” and recommend that future investigations realize the tensile strength analysis. Additionally, as indicated in the conclusions section, it is necessary to continue with this research to evaluate other properties (e.g., antimicrobial, anti-inflammatory, cytotoxic, among others) and be able to determine broadly the use of the films.
- The controlled release mechanism of as-prepared films should be discussed in-depth, which is also mentioned in the Introduction Section (from line 72). In the discussion part, authors can refer to the literatures that always have the detailed classification on this.
We appreciate the comment. In this part of the investigation, we only consider the release time of the active compounds through the solubilization of the films in artificial saliva (ranging from 9 –12 min). The solubilization of the films allows the complete release of the active compounds incorporated within the polymeric matrix toward the artificial saliva. This information was added in the abstract and discussion (3.2.2.2 Solubility tests in artificial saliva) sections.
- How are the durance of as-developed films? In other words, can the films be maintained long-term antioxidant activity in the presence of hash ambient conditions (e.g., water immersion, CO2 attack, …)
We appreciate the comment. The objective of this investigation was not to carry out tests of the useful life of the films. However, it is very important, of course, we will do it in future studies in our working group. At this time, we only consider evaluating the solubility of the films by immersion in artificial saliva since these materials were designed for applications in the oral cavity.
- The language of this manuscript should be polished thoroughly for trivial grammar and stylistic errors. Ok. The article was submitted for an English language review.

Reviewer 3 Report
The development of new food coatings, especially biodegradable coatings, is a very important trend in the food industry and polymer science. Perhaps some reviewers will say that there are many such publications. However, I am of a different opinion. Each such publication is valuable and brings us closer and closer to understanding the "composition-property" relationships, which are of great fundamental and applied importance in materials science. The article is well written and logical, well illustrated. The text of the article is clear and easy to understand. In general, a lot of positive. I recommend a minor revision before acceptance to improve the quality of the article. I ask the authors to describe the possibility of swelling of these films under the action of environmental moisture and in contact with food. Is the swelling pronounced or negligible?Language is fine.
Author Response
We appreciate the comments and suggestions of the reviewers regarding the manuscript entitled “Production and preliminary characterization of linseed mucilage-based films loaded with cardamom (Elettaria cardamomum), and copaiba (Copaifera officinalis)”. The suggestions and modifications indicated by the reviewers were made in the manuscript downloaded from the link using "track changes", the responses for the reviewers are shown below. Additionally, we clarify that before submission the manuscript was subjected to an English language review.
The development of new food coatings, especially biodegradable coatings, is a very important trend in the food industry and polymer science. Perhaps some reviewers will say that there are many such publications. However, I am of a different opinion. Each such publication is valuable and brings us closer and closer to understanding the "composition-property" relationships, which are of great fundamental and applied importance in materials science. The article is well written and logical, well illustrated. The text of the article is clear and easy to understand. In general, a lot of positive. I recommend a minor revision before acceptance to improve the quality of the article. I ask the authors to describe the possibility of swelling of these films under the action of environmental moisture and in contact with food. Is the swelling pronounced or negligible?
Comments on the Quality of English Language
Language is fine.
We appreciate the comments and suggestions of reviewer 3. Ok, information about film swelling was added in the text (section 3.2.2.2 Solubility tests in artificial saliva).

Round 2
Reviewer 1 Report
The authors address all suggestions recommended by me. Now, the article may be accepted.
Reviewer 2 Report
Authors have modified their manuscript; thus, it can be accepted.